# Fast Training of Pose Detectors in the Fourier Domain

João F. Henriques        Pedro Martins        Rui Caseiro        Jorge Batista

Institute of Systems and Robotics
University of Coimbra
{henriques,pedromartins,ruicaseiro,batista}@isr.uc.pt

## Abstract

In many datasets, the samples are related by a known image transformation, such as rotation, or a repeatable non-rigid deformation. This applies to both datasets with the same objects under different viewpoints, and datasets augmented with virtual samples. Such datasets possess a high degree of redundancy, because geometrically-induced transformations should preserve intrinsic properties of the objects. Likewise, ensembles of classifiers used for pose estimation should also share many characteristics, since they are related by a geometric transformation. By assuming that this transformation is norm-preserving and cyclic, we propose a closed-form solution in the Fourier domain that can eliminate most redundancies. It can leverage off-the-shelf solvers with no modification (e.g. libsvm), and train several pose classifiers simultaneously at no extra cost. Our experiments show that training a sliding-window object detector and pose estimator can be sped up by orders of magnitude, for transformations as diverse as planar rotation, the walking motion of pedestrians, and out-of-plane rotations of cars.

## 1   Introduction

To cope with the rich variety of transformations in natural images, recognition systems require a representative sample of possible variations. Some of those variations must be learned from data (e.g. non-rigid deformations), while others can be virtually generated (e.g. translation or rotation). Recently, there has been a renewed interest in augmenting datasets with virtual samples, both in the context of supervised [23, 17] and unsupervised learning [6]. This augmentation has the benefits of regularizing high-capacity classifiers [6], while learning the natural invariances of the visual world.

Some kinds of virtual samples can actually make learning easier – for example, with horizontally-flipped virtual samples [7, 4, 17], half of the weights of the template in the Dalal-Triggs detector [4] become redundant by horizontal symmetry. A number of very recent works [14, 13, 8, 1] have shown that cyclically translated virtual samples *also* constrain learning problems, which allows impressive gains in computational efficiency. The core of this technique relies on approximately diagonalizing the data matrix with the Discrete Fourier Transform (DFT).

In this work, we show that the "Fourier trick" is not unique to cyclic translation, but can be generalized to other cyclic transformations. Our model captures a wide range of useful image transformations, yet retains the ability to accelerate training with the DFT. As it is only implicit, we can accelerate training in both datasets of virtual samples and natural datasets with pose annotations.

Also due to the geometrically-induced structure of the training data, our algorithm can obtain several transformed pose classifiers simultaneously. Some of the best object detection and pose estimation systems currently learn classifiers for different poses independently [10, 7, 19], and we show how joint learning of these classifiers can dramatically reduce training times.

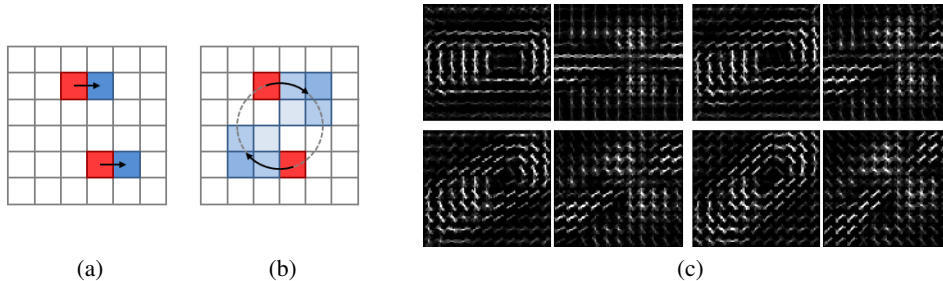

Figure 1: **(a)** The horizontal translation of a $6 \times 6$ image, by 1 pixel, can be achieved by a $36 \times 36$ permutation matrix $P$ that reorders elements appropriately (depicted is the reordering of 2 pixels). **(b)** Rotation by a fixed angle, with linearly-interpolated pixels, requires a more general matrix $Q$. By studying its influence on a dataset of rotated samples, we show how to accelerate learning in the Fourier domain. Our model can also deal with other transformations, including non-rigid. **(c)** Example HOG template (a car from the Google Earth dataset) at 4 rotations learned by our model. Positive weights are on the first and third column, others are negative.

## 1.1 Contributions

Our contributions are as follows: 1) We generalize a previous successful model for translation [14, 13] to other transformations, and analyze the properties of datasets with many transformed images; 2) We present closed-form solutions that fully exploit the known structure of these datasets, for Ridge Regression and Support Vector Regression, based on the DFT and off-the-shelf solvers; 3) With the same computational cost, we show how to train multiple classifiers for different poses simultaneously; 4) Since our formulas do not require explicitly estimating or knowing the transformation, we demonstrate applicability to both datasets of virtual samples and structured datasets with pose annotations. We achieve performance comparable to naive algorithms on 3 widely different tasks, while being several orders of magnitude faster.

## 1.2 Related work

There is a vast body of works on image transformations and invariances, of which we can only mention a few. Much of the earlier computer vision literature focused on finding viewpoint-invariant patterns [22]. They were based on image or scene-space coordinates, on which geometric transformations can be applied directly, however they do not apply to modern appearance-based representations. To relate complex transformations with appearance descriptors, a classic approach is to use tangent vectors [3, 26, 16], which represent a first-order approximation. However, the desire for more expressiveness has motivated the search for more general models.

Recent works have begun to approximate transformations as matrix-vector products, and try to estimate the transformation matrix explicitly. Tamaki et al. [27] do so for blur and affine transformations in the context of LDA, while Miao et al. [21] approximate affine transformations with an E-M algorithm, based on a Lie group formulation. They estimate a basis for the transformation operator or the transformed images, which is a hard analytical/inference problem in itself. The involved matrices are extremely large for moderately-sized images, necessitating dimensionality reduction techniques such as PCA, which may be suboptimal.

Several works focus on rotation alone [25, 18, 28, 2], most of them speeding up computations using Fourier analysis, but they all explicitly estimate a reduced basis on which to project the data. Another approach is to learn a transformation from data, using more parsimonious factored or deep models [20]. In contrast, our method generalizes to other transformations and avoids a potentially costly transformation model or basis estimation.

## 2 The cyclic orthogonal model for image transformations

Consider the $m \times 1$ vector $\mathbf{x}$, obtained by vectorizing an image, i.e. stacking its elements into a vector. The particular order does not matter, as long as it is consistent. The image may be a 3-

dimensional array that contains multiple channels, such as RGB, or the values of a densely-sampled image descriptor.

We wish to quickly train a classifier or regressor with transformed versions of sample images, to make it robust to those transformations. The model we will use is an $m \times m$ orthogonal matrix $Q$, which will represent an incremental transformation of an image as $Q\mathbf{x}$ (for example, a small translation or rotation, see Fig. 1-a and 1-b). We can traverse different poses w.r.t. that transformation, $p \in \mathbb{Z}$, by repeated application of $Q$ with a matrix power, $Q^p\mathbf{x}$.

In order for the number of poses to be finite, we must require the transformation to be *cyclic*, $Q^s = Q^0 = I$, with some period $s$. This allows us to store all versions of $\mathbf{x}$ transformed to different poses as the rows of an $s \times m$ matrix,

$$C_Q(\mathbf{x}) = \begin{bmatrix} \left(Q^0\mathbf{x}\right)^T \\ \left(Q^1\mathbf{x}\right)^T \\ \vdots \\ \left(Q^{s-1}\mathbf{x}\right)^T \end{bmatrix} \tag{1}$$

Due to $Q$ being cyclic, any pose $p \in \mathbb{Z}$ can be found in the row $(p \bmod s) + 1$. Note that the first row of $C_Q(\mathbf{x})$ contains the untransformed image $\mathbf{x}$, since $Q^0$ is the identity $I$. For the purposes of training a classifier, $C_Q(\mathbf{x})$ can be seen as a data matrix, with one sample per row.

Although conceptually simple, we will show through experiments that this model can accurately capture a variety of natural transformations (Section 5.2). More importantly, we will show that $Q$ *never has to be created explicitly*. The algorithms we develop will be entirely data-driven, using an implicit description of $Q$ from a structured dataset, either composed of virtual samples (e.g., by image rotation), or natural samples (e.g. using pose annotations).

## 2.1 Image translation as a special case

A particular case of $Q$, and indeed what inspired the generalization that we propose, is the $s \times s$ cyclic shift matrix

$$P = \begin{bmatrix} 0_{s-1}^T & 1 \\ I_{s-1} & 0_{s-1} \end{bmatrix}, \tag{2}$$

where $0_{s-1}$ is an $(s-1) \times 1$ vector of zeros. This matrix cyclically permutes the elements of a vector $\mathbf{x}$ as $(x_1, x_2, x_3, \ldots, x_s) \rightarrow (x_s, x_1, x_2, \ldots, x_{s-1})$. If $\mathbf{x}$ is a one-dimensional horizontal image, with a single channel, then it is translated to the right by one pixel. An illustration is shown in Fig. 1-a. By exploiting its relationship with the Discrete Fourier Transform (DFT), the cyclic shift model has been used to accelerate a variety of learning algorithms in computer vision [14, 13, 15, 8, 1], with suitable extensions to 2D and multiple channels.

## 2.2 Circulant matrices and the Discrete Fourier Transform

The basis for this optimization is the fact that the data matrix $C_P(\mathbf{x})$, or $C(\mathbf{x})$ for short, formed by all cyclic shifts of a sample image $\mathbf{x}$, is *circulant* [5]. All circulant matrices are diagonalized by the DFT, which can be expressed as the eigendecomposition

$$C(\mathbf{x}) = U \operatorname{diag}\left(\mathcal{F}(\mathbf{x})\right) U^H, \tag{3}$$

where $.^H$ is the Hermitian transpose (i.e., transposition and complex-conjugation), $\mathcal{F}(\mathbf{x})$ denotes the DFT of a vector $\mathbf{x}$, and $U$ is the unitary DFT basis. The constant matrix $U$ can be used to compute the DFT of any vector, since it satisfies $U\mathbf{x} = \frac{1}{\sqrt{s}}\mathcal{F}(\mathbf{x})$. This is possible due to the linearity of the DFT, though in practice the Fast Fourier Transform (FFT) algorithm is used instead. Note that $U$ is symmetric, $U^T = U$, and unitary, $U^H = U^{-1}$. When working in Fourier-space, Eq. 3 shows that circulant matrices in a learning problem become diagonal, which drastically reduces the needed computations. For multiple channels or more images, they may become block-diagonal, but the principles remain the same [13].

An important open question was whether the same diagonalization trick can be applied to image transformations other than translation. We will show that this is true, using the model from Eq. 1.

## 3 Fast training with transformations of a single image

We will now focus on the main derivations of our paper, which allow us to quickly train a classifier with virtual samples generated from an image $\mathbf{x}$ by repeated application of the transformation $Q$. This section assumes only a single image $\mathbf{x}$ is given for training, which makes the presentation simpler and we hope will give valuable insight into the core of the technique. Section 4 will expand it to full generality, with training sets of an arbitrary number of images, all transformed by $Q$.

The first step is to show that some aspect of the data is diagonalizable by the DFT, which we do in the following theorem.

**Theorem 1.** *Given an orthogonal cyclic matrix $Q$, i.e. satisfying $Q^T = Q^{-1}$ and $Q^s = Q^0$, then the $s \times m$ matrix $X = C_Q(\mathbf{x})$ (from Eq. 1) verifies the following:*

- The data matrix $X$ and the uncentered covariance matrix $X^H X$ are not circulant in general, unless $Q = P$ (from Eq. 2).
- The Gram matrix $G = XX^H$ is always circulant.

*Proof.* See Appendix A.1. □

Theorem 1 implies that the learning problem in its original form is not diagonalizable by the DFT basis. However, the same diagonalization is possible for the dual problem, defined by the Gram matrix $G$.

Because $G$ is circulant, it has only $s$ degrees of freedom and is fully specified by its first row $\mathbf{g}$ [11], $G = C(\mathbf{g})$. By direct computation from Eq. 1, we can verify that the elements of the first row $\mathbf{g}$ are given by $g_p = \mathbf{x}^T Q^{p-1} \mathbf{x}$. One interpretation is that $\mathbf{g}$ contains the auto-correlation of $\mathbf{x}$ through pose-space, i.e., the inner-product of $\mathbf{x}$ with itself as the transformation $Q$ is applied repeatedly.

### 3.1 Dual Ridge Regression

For now we will restrict our attention to Ridge Regression (RR), since it has the appealing property of having a solution in closed form, which we can easily manipulate. Section 4.1 will show how to extend these results to Support Vector Regression. The goal of RR is to find the linear function $f(\mathbf{x}) = \mathbf{w}^T \mathbf{x}$ that minimizes a regularized squared error: $\sum_i \left( f(\mathbf{x}_i) - y_i \right)^2 + \lambda \left\| \mathbf{w} \right\|^2$.

Since we have $s$ samples in the data matrix under consideration (Eq. 1), there are $s$ dual variables, stored in a vector $\boldsymbol{\alpha}$. The RR solution is given by $\boldsymbol{\alpha} = (G + \lambda I)^{-1} \mathbf{y}$ [24], where $G = XX^H$ is the $s \times s$ Gram matrix, $\mathbf{y}$ is the vector of $s$ labels (one per pose), and $\lambda$ is the regularization parameter. The dual form of RR is usually associated with non-linear kernels [24], but since this is not our case we can compute the explicit primal solution with $\mathbf{w} = X^T \boldsymbol{\alpha}$, yielding

$$\mathbf{w} = X^T \left( G + \lambda I \right)^{-1} \mathbf{y}. \tag{4}$$

Applying the circulant eigendecomposition (Eq. 3) to $G$, and substituting it in Eq. 4,

$$\mathbf{w} = X^T \left( U \operatorname{diag}\left( \hat{\mathbf{g}} \right) U^H + \lambda U U^H \right)^{-1} \mathbf{y} = X^T U \left( \operatorname{diag}\left( \hat{\mathbf{g}} + \lambda \right) \right)^{-1} U^H \mathbf{y}, \tag{5}$$

where we introduce the shorthand $\hat{\mathbf{g}} = \mathcal{F}\left( \mathbf{g} \right)$, and similarly $\hat{\mathbf{y}} = \mathcal{F}\left( \mathbf{y} \right)$. Since inversion of a diagonal matrix can be done element-wise, and its multiplication by the vector $U^H \mathbf{y}$ amounts to an element-wise product, we obtain

$$\mathbf{w} = X^T \mathcal{F}^{-1} \left( \frac{\hat{\mathbf{y}}}{\hat{\mathbf{g}} + \lambda} \right), \tag{6}$$

where $\mathcal{F}^{-1}$ denotes the inverse DFT, and the division is taken element-wise. This formula allows us to replace a costly matrix inversion with fast DFT and element-wise operations. We also do not need to compute and store the full $G$, as the auto-correlation vector $\mathbf{g}$ suffices. As we will see in the next section, there is a simple modification to Eq. 6 that turns out to be very useful for pose estimation.

## 3.2 Training several components simultaneously

A relatively straightforward way to estimate the object pose in an input image $\mathbf{x}$ is to train a classifier for each pose (which we call *components*), evaluate all of them and take the maximum, i.e.

$$f_{\text{pose}}(\mathbf{x}) = \arg\max_p \mathbf{w}_p^T \mathbf{x}. \tag{7}$$

This can also be used as the basis for a pose-invariant classifier, by replacing argmax with max [10]. Of course, training one component per pose can quickly become expensive. However, we can exploit the fact that these training problems become tightly related when the training set contains transformed images.

Recall that $\mathbf{y}$ specifies the labels for a training set of $s$ transformed images, one label per pose. Without any loss of generality, suppose that the label is 1 for a given pose $t$ and 0 for all others, i.e. $\mathbf{y}$ contains a single peak at element $t$. Then by shifting the peak with $P^p\mathbf{y}$, we will train a classifier for pose $t + p$. In this manner we can train classifiers for all poses simply by varying the labels $P^p\mathbf{y}$, with $p = 0, \ldots, s - 1$.

Based on Eq. 6, we can concatenate the solutions for all $s$ components into a single $m \times s$ matrix,

$$W = \left[ \ \mathbf{w}_0 \ \middle| \ \cdots \ \middle| \ \mathbf{w}_{s-1} \ \right] = X^T (G + \lambda I)^{-1} \left[ \ P^0\mathbf{y} \ \middle| \ \cdots \ \middle| \ P^{s-1}\mathbf{y} \ \right] \tag{8}$$

$$= X^T (G + \lambda I)^{-1} C^T (\mathbf{y}). \tag{9}$$

Diagonalization yields

$$W^T = \mathcal{F}^{-1} \left( \text{diag} \left( \frac{\hat{\mathbf{y}}^*}{\hat{\mathbf{g}} + \lambda} \right) \mathcal{F}(X) \right), \tag{10}$$

where .* denotes complex-conjugation. Since their arguments are matrices, the DFT/IDFT operations here work along each column. The product of $\mathcal{F}(X)$ by the diagonal matrix simply amounts to multiplying each of its rows by a scalar factor, which is inexpensive. Eq. 10 has nearly the same computational cost as Eq. 6, which trains a single classifier.

## 4 Transformation of multiple images

The training method described in the previous section would find little applicability for modern recognition tasks if it remained limited to transformations of a single image. Naturally, we would like to use $n$ images $\mathbf{x}_i$. We now have a dataset of $ns$ samples, which can be divided into $n$ sample groups $\{Q^{p-1}\mathbf{x}_i | p = 1, \ldots, s\}$, each containing the transformed versions of one image.

This case becomes somewhat complicated by the fact that the data matrix $X$ now has three dimensions – the $m$ features, the $n$ sample groups, and the $s$ poses of each sample group. In this $m \times n \times s$ array, each column vector (along the first dimension) is defined as

$$X_{\bullet ip} = Q^{p-1}\mathbf{x}_i, \qquad\qquad i = 1, \ldots, n; \ p = 1, \ldots, s, \tag{11}$$

where we have used $\bullet$ to denote a one-dimensional slice of the three-dimensional array $X$.[1] A two-dimensional slice will be denoted by $X_{\bullet\bullet p}$, which yields a $m \times n$ matrix, one for each $p = 1, \ldots, s$.

Through a series of block-diagonalizations and reorderings, we can show (Appendix A.2-A.5) that the solution $W$, of size $m \times s$, describing all $s$ components (similarly to Eq. 10), is obtained with

$$\hat{W}_{\bullet p} = \hat{X}_{\bullet\bullet p} (\hat{\mathbf{g}}_{\bullet\bullet p} + \lambda I)^{-1} \hat{Y}_{\bullet p}^*, \qquad\qquad p = 1, \ldots, s, \tag{12}$$

where a hat ˆ over an array denotes the DFT along the dimension that has size $s$ (e.g. $\hat{X}$ is the DFT of $X$ along the third dimension), $Y_{ip}$ specifies the label of the sample with pose $p$ in group $i$, and $\mathbf{g}$ is the $n \times n \times s$ array with elements

$$\mathbf{g}_{ijp} = \mathbf{x}_i^T Q^{p-1} \mathbf{x}_j = X_{\bullet i1}^T X_{\bullet jp}, \qquad\qquad i,j = 1, \ldots, n; \; p = 1, \ldots, s. \qquad (13)$$

It may come as a surprise that, after all these changes, Eq. 12 still essentially looks like a dual Ridge Regression (RR) problem (compare it to Eq. 4). Eq. 12 can be interpreted as splitting the original problem into $s$ smaller problems, one for each Fourier frequency, which are independent and can be solved in parallel. A Matlab implementation is given in Appendix B.[2]

## 4.1 Support Vector Regression

Given that we can decompose such a large RR problem into $s$ smaller RR problems, by applying the DFT and slicing operators (Eq. 12), it is natural to ask whether the same can be done with other algorithms. Leveraging a recent result [13], where this was done for image translation, the same steps can be repeated for the dual formulation of other algorithms, such as Support Vector Regression (SVR). Although RR can deal with complex data, SVR requires an extension to the complex domain, which we show in Appendix A.6. We give a Matlab implementation in Appendix B, which can use any off-the-shelf SVR solver without modification.

## 4.2 Efficiency

Naively training one detector per pose would require solving $s$ large $ns \times ns$ systems (either with RR or SVR). In contrast, our method learns *jointly* all detectors using $s$ much smaller $n \times n$ subproblems. The computational savings can be several orders of magnitude for large $s$. Our experiments seem to validate this conclusion, even in relatively large recognition tasks (Section 6).

# 5 Orthogonal transformations in practice

Until now, we avoided the question of how to compute a transformation model $Q$. This may seem like a computational burden, not to mention a hard estimation problem – for example, what is the cyclic orthogonal matrix $Q$ that models planar rotations with period $s$? Inspecting Eq. 12-13, however, reveals that we do not need to form $Q$ explicitly, but can work with just a data matrix $X$ of transformed images. From there on, we exploit the knowledge that this data was obtained from *some* matrix $Q$, and that is enough to allow fast training in the Fourier domain. This allows a great deal of flexibility in implementation.

## 5.1 Virtual transformations

One way to obtain a structured data matrix $X$ is with virtual samples. From the original dataset of $n$ samples, we can generate $ns$ virtual samples using a standard image operator (e.g. planar rotation). However, we should keep in mind that the accuracy of the proposed method will be affected by how much the image operator resembles a pure cyclic orthogonal transformation.

**Linearity.** Many common image transformations, such as rotation or scale, are implemented by nearest-neighbor or bilinear interpolation. For a fixed amount of rotation or scale, these functions are linear functions in the input pixels, i.e. each output pixel is a fixed linear combination of some of the input pixels. As such, they fulfill the linearity requirement.

**Orthogonality.** For an operator to be orthogonal, it must preserve the $L^2$ norm of its inputs. At the expense of introducing some non-linearity, we simply renormalize each virtual sample to have the same norm as the original sample, which seems to work well in practice (Section 6).

**Cyclicity.** We conducted some experiments with planar rotation on satellite imagery (Section 6.1) – rotation by $360/s$ degrees is cyclic with period $s$. In the future, we plan to experiment with non-cyclic operators (similar to how cyclic translation is used to approximate image translation [14]).

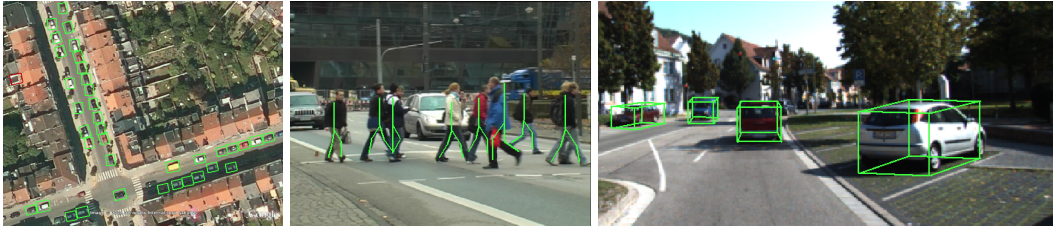

Figure 2: Example detections and estimated poses in 3 different settings. We can accelerate training with (a) planar rotations (Google Earth), (b) non-rigid deformations in walking pedestrians (TUD-Campus/TUD-Crossing), and (c) out-of-plane rotations (KITTI). Best viewed in color.

## 5.2 Natural transformations

Another interesting possibility is to use pose annotations to create a structured data matrix. This data-driven approach allows us to consider more complicated transformations than those associated with virtual samples. Given $s$ views of $n$ objects under different poses, we can build the $m \times n \times s$ data matrix $X$ and use the same methodology as before. In Section 6 we describe experiments with the walk cycle of pedestrians, and out-of-plane rotations of cars in street scenes. These transformations are cyclic, though highly non-linear, and we use the same renormalization as in Section 5.1.

## 5.3 Negative samples

One subtle aspect is how to obtain a structured data matrix from negative samples. This is simple for virtual transformations, but not for natural transformations. For example, with planar rotation we can easily generate rotated negative samples with arbitrary poses. However, the same operation with walk cycles of pedestrians is not defined. How do we advance the walk cycle of a non-pedestrian? As a pragmatic solution, we consider that negative samples are unaffected by natural transformations, so a negative sample is constant for all $s$ poses. Because the DFT of a constant signal is 0, except for the DC value (the first frequency), we can ignore untransformed negative samples in all subproblems for $p \neq 1$ (Eq. 12). This simple observation can result in significant computational savings.

## 6 Experiments

To demonstrate the generality of the proposed model, we conducted object detection and pose estimation experiments on 3 widely different settings, which will be described shortly. We implemented a detector based on Histogram of Oriented Gradients (HOG) templates [4] with multiple components [7]. This framework forms the basis on which several recent advances in object detection are built [19, 10, 7]. The baseline algorithm independently trains $s$ classifiers (components), one per pose, enabling pose-invariant object detection and pose prediction (Eq. 7). Components are then calibrated, as usual for detectors with multiple components [7, 19]. The proposed method does not require any ad-hoc calibration, since the components are jointly trained and related by the orthogonal matrix $Q$, which preserves their $L^2$ norm.

For the performance evaluation, ground truth objects are assigned to hypothesis by the widely used Pascal criterion of bounding box overlap [7]. We then measure average precision (AP) and pose error (as $e_{\text{pose}}/s$, where $e_{\text{pose}}$ is the discretized pose difference, taking wrap-around into account). We tested two variants of each method, trained with both RR and SVR. Although parallelization is trivial, we report timings for single-core implementations, which more accurately reflect the total CPU load. As noted in previous work [13], detectors trained with SVR have very similar performance to those trained with Support Vector Machines.

## 6.1 Planar rotation in satellite images (Google Earth)

Our first test will be on a car detection task on satellite imagery [12], which has been used in several works that deal with planar rotation [25, 18]. We annotated the orientations of 697 objects over half the 30 images of the dataset. The first 7 annotated images were used for training, and the remaining 8 for validation. We created a structured data matrix $X$ by augmenting each sample with 30 virtual

|  |  | Google Earth | | | TUD Campus/Crossing | | | KITTI | | |
|---|---|---|---|---|---|---|---|---|---|---|
|  |  | Time (s) | AP | Pose | Time (s) | AP | Pose | Time (s) | AP | Pose |
| **Fourier training** | SVR | **4.5** | 73.0 | 9.4 | **0.1** | 81.5 | 9.3 | **15.0** | 53.5 | 14.9 |
|  | RR | **3.7** | 71.4 | 10.0 | **0.08** | 82.2 | 8.9 | **15.5** | 53.4 | 15.0 |
| Standard | SVR | 130.7 | 73.2 | 9.8 | 40.5 | 80.2 | 9.5 | 454.2 | 56.5 | 13.8 |
|  | RR | 399.3 | 72.7 | 10.3 | 45.8 | 81.6 | 9.4 | 229.6 | 54.5 | 14.0 |

Table 1: Results for pose detectors trained with Support Vector Regression (SVR) and Ridge Regression (RR). We report training time, Average Precision (AP) and pose error (both in percentage).

samples, using 12º rotations. A visualization of trained weights is shown in Fig. 1-c and Appendix B. Experimental results are presented in Table 1. Recall that our primary goal is to demonstrate faster training, *not* to improve detection performance, which is reflected in the results. Nevertheless, the two proposed fast Fourier algorithms are 29 to 107$\times$ faster than the baseline algorithms.

### 6.2 Walk cycle of pedestrians (TUD-Campus and TUD-Crossing)

We can consider a walking pedestrian to undergo a cyclic non-rigid deformation, with each period corresponding to one step. Because this transformation is time-dependent, we can learn it from video data. We used TUD-Campus for training and TUD-Crossing for testing (see Fig. 2). We annotated a key pose in all 272 frames, so that the images of a pedestrian between two key poses represent a whole walk cycle. Sampling 10 images per walk cycle (corresponding to 10 poses), we obtained 10 sample groups for training, for a total of 100 samples.

From Table 1, the proposed algorithms seem to slightly outperform the baseline, showing that these non-rigid deformations can be accurately accounted for. However, they are over 2 orders of magnitude faster. In addition to the speed benefits observed in Section 6.1, another factor at play is that for natural transformations we can ignore the negative samples in $s-1$ of the subproblems (Section 5.3), whereas the baseline algorithms must consider them when training each of the $s$ components.

### 6.3 Out-of-plane rotations of cars in street scenes (KITTI)

For our final experiment, we will attempt to demonstrate that the speed advantage of our method still holds for difficult out-of-plane rotations. We chose the very recent KITTI benchmark [9], which includes an object detection set of 7481 images of street scenes. The facing angle of cars (along the vertical axis) is provided, which we bin into 15 discrete poses. We performed an 80-20% train-test split of the images, considering cars of "moderate" difficulty [9], and obtained 73 sample groups for training with 15 poses each (for a total of 1095 samples).

Table 1 shows that the proposed method achieves competitive performance, but with a dramatically lower computational cost. The results agree with the intuition that out-of-plane rotations strain the assumptions of linearity and orthogonality, since they result in large deformations of the object. Nevertheless, the ability to learn a useful model under such adverse conditions shows great promise.

## 7 Conclusions and future work

In this work, we derived new closed-form formulas to quickly train several pose classifiers at once, and take advantage of the structure in datasets with pose annotation or virtual samples. Our implicit transformation model seems to be surprisingly expressive, and in future work we would like to experiment with other transformations, including non-cyclic. Other interesting directions include larger-scale variants and the composition of multiple transformations.

**Acknowledgements.** The authors would like to thank João Carreira for valuable discussions. They also acknowledge support by the FCT project PTDC/EEA-CRO/122812/2010, grants SFRH/BD75459/2010, SFRH/BD74152/2010, and SFRH/BPD/90200/2012.

## Footnotes

[1]For reference, our slice notation $\bullet$ works the same way as the slice notation : in Matlab or NumPy.

[2]The supplemental material is available at: `www.isr.uc.pt/~henriques/transformations/`

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
