[Supplementary Material 1]

# Fast Training of Pose Detectors in the Fourier Domain

**João F. Henriques**      **Pedro Martins**      **Rui Caseiro**      **Jorge Batista**

Institute of Systems and Robotics

University of Coimbra

{henriques,pedromartins,ruicaseiro,batista}@isr.uc.pt

## A    Proofs

This appendix contains proofs that were not included in the main paper to meet length requirements. See Appendix B for MATLAB code and additional figures.

### A.1    Proof of Theorem 1

Though the main claim of the Theorem is the last one, there are also other claims which we will prove in turn.

*The data matrix $X$ and the uncentered covariance matrix $X^H X$ are not circulant in general.*

This can be demonstrated with a simple counterexample. Consider the following transformation $Q$, which simply reverses the order of any $3 \times 1$ input vector:

$$Q = \begin{bmatrix} 0 & 0 & 1 \\ 0 & 1 & 0 \\ 1 & 0 & 0 \end{bmatrix} \tag{A.1}$$

It is orthogonal and cyclic with period $s = 2$ ($Q^2 = Q^0 = I$).

The corresponding data matrix (Eq. 1) is not square, and thus cannot be circulant [1]

$$X = \begin{bmatrix} x_1 & x_2 & x_3 \\ x_3 & x_2 & x_1 \end{bmatrix} \tag{A.2}$$

Another necessary (but not sufficient) condition for a matrix to be circulant is that its diagonal elements are constant [1]. By direct computation, the uncentered covariance matrix $X^H X$ fails this requirement; its diagonal elements are given by $\left[ x_1^2 + x_3^2, \ 2x_2^2, \ x_1^2 + x_3^2 \right]$.

*The data matrix $X$ and the uncentered covariance matrix $X^H X$ are circulant for $Q = P$.*

This is an earlier result [2].

*The Gram matrix $G = X X^H$ is circulant.*

We have

$$G_{pr} = (Q^p \mathbf{x})^T Q^r \mathbf{x} = \mathbf{x}^T (Q^p)^{-1} Q^r \mathbf{x} = \mathbf{x}^T Q^{r-p} \mathbf{x} = \mathbf{x}^T Q^{(r-p) \bmod s} \mathbf{x}, \tag{A.3}$$

where the second equivalence is due to orthogonality, and the last one is due to cyclicity. The strict dependence on $(r - p) \bmod s$ implies that $G$ is circulant [1].

## A.2 Fast solution in the dual for training with multiple transformed images

We are given $n$ sample groups, each of them containing $s$ transformed versions of an image $\mathbf{x}_i$. Let us organize the data into $n$ blocks $X(i)$, one per sample group, each block with size $s \times m$:

$$X(i) = C_Q(\mathbf{x}_i), \qquad\qquad i = 1, \ldots, n \qquad\qquad \text{(A.4)}$$

The full $ns \times m$ data matrix is obtained by vertical concatenation of all the $X(i)$. We can compute the corresponding Gram matrix easily since it is just the product of two block matrices. It is composed of $n^2$ blocks, each one of size $s \times s$, defined by

$$G(i, j) = X^T(i)\, X(j), \qquad\qquad i, j = 1, \ldots, n. \qquad\qquad \text{(A.5)}$$

The Ridge Regression (RR) problem with an $ns \times ns$ Gram matrix composed of these blocks is given by

$$\begin{bmatrix} \boldsymbol{\alpha}(1) \\ \vdots \\ \boldsymbol{\alpha}(n) \end{bmatrix} = \left( \begin{bmatrix} G(1,1) & \cdots & G(1,n) \\ \vdots & \ddots & \vdots \\ G(n,1) & \cdots & G(n,n) \end{bmatrix} + \lambda I \right)^{-1} \begin{bmatrix} \mathbf{y}(1) \\ \vdots \\ \mathbf{y}(n) \end{bmatrix}, \qquad \text{(A.6)}$$

where $\boldsymbol{\alpha}(i)$ are $s \times 1$ vectors of solution coefficients, and $\mathbf{y}(i)$ are $s \times 1$ vectors of target labels.

Each block $G(i, j)$ verifies Theorem 1, which means that they are circulant. As such, they are defined by their first row,

$$G(i, j) = C\left( \mathbf{x}_i\, C_Q^T(\mathbf{x}_j) \right). \qquad\qquad \text{(A.7)}$$

We can diagonalize the blocks of the Gram matrix individually, by transforming the problem (block-wise) to the Fourier domain. Eq. A.6 is equivalent to

$$\begin{bmatrix} \hat{\boldsymbol{\alpha}}(1) \\ \vdots \\ \hat{\boldsymbol{\alpha}}(n) \end{bmatrix} = \left( \begin{bmatrix} \hat{G}(1,1) & \cdots & \hat{G}(1,n) \\ \vdots & \ddots & \vdots \\ \hat{G}(n,1) & \cdots & \hat{G}(n,n) \end{bmatrix} + \lambda I \right)^{-1} \begin{bmatrix} \hat{\mathbf{y}}(1) \\ \vdots \\ \hat{\mathbf{y}}(n) \end{bmatrix}, \qquad \text{(A.8)}$$

with the Fourier-domain variables $\hat{\boldsymbol{\alpha}}(i) = U\boldsymbol{\alpha}(i)$, $\hat{\mathbf{y}}(i) = U\mathbf{y}(i)$, and

$$\hat{G}(i, j) = U^H G(i, j) U, \qquad\qquad i, j = 1, \ldots, n, \qquad\qquad \text{(A.9)}$$

The identity $I$ is unaffected by $U$ because the later is unitary.

Since $G(i, j)$ is circulant, $\hat{G}(i, j)$ must be diagonal, i.e.,

$$\hat{G}_{pr}(i, j) = 0, \text{ if } p \neq r. \qquad\qquad \text{(A.10)}$$

We can turn the Gram matrix with diagonal blocks into a block-diagonal matrix by a permutation of its rows and columns. Define $s^2$ blocks, each one $n \times n$, with elements obtained just by reordering the elements of $\hat{G}(i, j)$:

$$G'_{ij}(p, r) = \hat{G}_{pr}(i, j), \qquad\qquad i, j = 1, \ldots, n. \qquad\qquad \text{(A.11)}$$

The two forms offer different views into the same data. $\hat{G}(i, j)$ describes the interactions through pose-space, after fixing two samples $i$ and $j$. $G'(p, r)$ emphasizes the interactions between pairs of samples, for a given Fourier frequency.

Given Eq. A.10 and Eq. A.11, we know that the off-diagonal $G'(p,r)$ blocks must be zero, i.e.,

$$G'(p,r) = \mathbf{0}, \text{ if } p \neq r, \tag{A.12}$$

with $\mathbf{0}$ denoting an $n \times n$ matrix of zeros. The RR problem in the permuted domain is then

$$\begin{bmatrix} \boldsymbol{\alpha}'(1) \\ \boldsymbol{\alpha}'(2) \\ \vdots \\ \boldsymbol{\alpha}'(s) \end{bmatrix} = \left( \begin{bmatrix} G'(1,1) & \mathbf{0} & \cdots & \mathbf{0} \\ \mathbf{0} & G'(2,2) & \cdots & \mathbf{0} \\ \vdots & \vdots & \ddots & \vdots \\ \mathbf{0} & \mathbf{0} & \cdots & G'(s,s) \end{bmatrix} + \lambda I \right)^{-1} \begin{bmatrix} \mathbf{y}'(1) \\ \mathbf{y}'(2) \\ \vdots \\ \mathbf{y}'(s) \end{bmatrix}, \tag{A.13}$$

where $\boldsymbol{\alpha}'_i(p) = \hat{\boldsymbol{\alpha}}_p(i)$ and $\mathbf{y}'_i(p) = \hat{\mathbf{y}}_p(i)$ are the remaining variables under the same permutation.

By direct computation with the rules of block matrices, we obtain

$$\begin{bmatrix} \boldsymbol{\alpha}'(1) \\ \boldsymbol{\alpha}'(2) \\ \vdots \\ \boldsymbol{\alpha}'(s) \end{bmatrix} = \begin{bmatrix} (G'(1,1) + \lambda I)^{-1} \mathbf{y}'(1) \\ (G'(2,2) + \lambda I)^{-1} \mathbf{y}'(2) \\ \vdots \\ (G'(s,s) + \lambda I)^{-1} \mathbf{y}'(s) \end{bmatrix}, \tag{A.14}$$

or more concisely,

$$\boldsymbol{\alpha}'(p) = (G'(p,p) + \lambda I)^{-1} \mathbf{y}'(p), \qquad p = 1, \ldots, s. \tag{A.15}$$

Note that Eq. A.15 hinges on the earlier definitions of $\boldsymbol{\alpha}'(p)$, $G'(p,p)$ and $\mathbf{y}'(p)$, which are Fourier-transformed and permuted versions of the original quantities.

### A.3 Formulation using multi-dimensional arrays

To make Eq. A.15 more self-contained, we can express it using multi-dimensional arrays, by tracing back the elements of $\boldsymbol{\alpha}'(p)$, $G'(p,p)$ and $\mathbf{y}'(p)$.

Define the $n \times n \times s$ array of unique inner-products $\mathbf{g}$, with elements

$$\mathbf{g}_{ijp} = \mathbf{x}_i^T Q^{p-1} \mathbf{x}_j. \tag{A.16}$$

Also, define the $n \times s$ matrix $Y$, where the element $Y_{ip}$ is the label of sample image $i$ for pose $p$.

Then Eq. A.15 can be implemented by taking the DFT of $Y$ along the second dimension and the DFT of $\mathbf{g}$ along the third dimension, i.e.,

$$\hat{Y} = \mathcal{F}_{(2)}(Y) \tag{A.17}$$
$$\hat{\mathbf{g}} = \mathcal{F}_{(3)}(\mathbf{g}), \tag{A.18}$$

and computing the $n \times s$ solution in the Fourier domain, $\hat{A}$, with

$$\hat{A}_{\bullet p} = (\hat{\mathbf{g}}_{\bullet\bullet p} + \lambda I)^{-1} \hat{Y}_{\bullet p}, \qquad p = 1, \ldots, s, \tag{A.19}$$

where $\hat{A}_{\bullet p}$ denotes the $p$th column from $\hat{A}$ (and similarly for $\hat{Y}$), while $\hat{\mathbf{g}}_{\bullet\bullet p}$ slices the $p$th subarray (of size $n \times n$) along the third dimension of $\hat{\mathbf{g}}$. For reference, the slicing operator $\bullet$ works the same way as the slicing operator $:$ in Matlab or NumPy.

Note that Eq. A.19 and Eq. A.15 are exactly the same, except with different notation.

We can retrieve the solution from Fourier space by taking the IDFT of $\hat{A}$ along the second dimension,

$$A = \mathcal{F}_{(2)}^{-1}\left(\hat{A}\right). \tag{A.20}$$

The element $A_{ip}$ is the dual coefficient of sample image $i$ for pose $p$.

## A.4 Solution for a single classifer

Using the data matrix in Eq. A.4 and the solution in the dual from Eq. A.19,

$$\mathbf{w} = \sum_{i=1}^{n} X^T(i)\, A_{i\bullet}\,. \tag{A.21}$$

## A.5 Solution for multiple pose classifiers

For multiple pose classifiers, we have

$$
\begin{aligned}
W = \left[\ \mathbf{w}_0\ \middle|\ \cdots\ \middle|\ \mathbf{w}_{s-1}\ \right] &= \sum_{i=1}^{n} X^T(i)\left[\ P^0 A_{i\bullet}\ \middle|\ \cdots\ \middle|\ P^{s-1}A_{i\bullet}\ \right] \\
&= \sum_{i=1}^{n} X^T(i) C^T\left(A_{i\bullet}\right),
\end{aligned}
\tag{A.22}
$$

because permuting the rows of the labels $Y$ results in the same permutation being applied to the rows of the solution $A$. Diagonalizing with $U$, we obtain

$$W^T = \mathcal{F}^{-1}\left(\sum_{i=1}^{n}\operatorname{diag}\left(\hat{A}_{i\bullet}^*\right)\mathcal{F}\left(X(i)\right)\right), \tag{A.23}$$

where $^*$ denotes complex-conjugation. Note that a product by a diagonal matrix on the left simply amounts to multiplying each row with one of the diagonal elements.

If $\hat{X}$ is the $m \times n \times s$ data matrix, Fourier-transformed in the third dimension, we can rewrite Eq. A.23 as

$$
\begin{aligned}
\hat{W}_{\bullet p} &= \hat{X}_{\bullet\bullet p}\hat{A}_{\bullet p}^* \\
&= \hat{X}_{\bullet\bullet p}\left(\hat{\mathbf{g}}_{\bullet\bullet p} + \lambda I\right)^{-1}\hat{Y}_{\bullet p}^*
\end{aligned}
$$

for $p = 1,\ldots,s$, and recover $W$ by taking the IDFT over the second dimension.

## A.6 Complex-valued Support Vector Regression in the Dual

We build on the primal solution for complex-valued Support Vector Regression (SVR) given in [2] (complex-SVR for short). We will restate it briefly. The standard $L^2$-SVR, with squared $\epsilon$-insensitive loss $\left|\mathbf{w}^H\mathbf{x}_j - y_j\right|_\epsilon = \max\left(0, \left|\mathbf{w}^H\mathbf{x}_j - y_j\right| - \epsilon\right)^2$, amounts to the following optimization problem:

$$\min_{\mathbf{w}}\|\mathbf{w}\|^2 + \frac{1}{\lambda}\sum_{j=1}^{n}\left|\mathbf{w}^H\mathbf{x}_j - y_j\right|_\epsilon. \tag{A.24}$$

It can be adapted for problems in the complex domain by defining the extended loss function [2],

$$\left|\mathbf{w}^H\mathbf{x} - y\right|_\epsilon = \left|\operatorname{Re}\left(\mathbf{w}^H\mathbf{x} - y\right)\right|_\epsilon + \left|\operatorname{Im}\left(\mathbf{w}^H\mathbf{x} - y\right)\right|_\epsilon, \tag{A.25}$$

where $\mathrm{Re}\left(\cdot\right)$ and $\mathrm{Im}\left(\cdot\right)$ denote the real and imaginary parts of a complex number, respectively. This complex-SVR can be shown [2] to be equivalent to a real-valued SVR with the following change of variables:

$$
\begin{aligned}
X' &= \left[ \begin{array}{cc} X_R & X_I \\ X_I & -X_R \end{array} \right] \\
\mathbf{y}' &= \left[ \begin{array}{c} \mathbf{y}_R \\ \mathbf{y}_I \end{array} \right] \\
\mathbf{w}' &= \left[ \begin{array}{c} \mathbf{w}_R \\ \mathbf{w}_I \end{array} \right],
\end{aligned}
\tag{A.26}
$$

where, for conciseness, the subscripts $R$ and $I$ denote real and imaginary parts, respectively (e.g., $X_R = \mathrm{Re}\left(X\right)$ and $X_I = \mathrm{Im}\left(X\right)$).

The algorithm that we propose in this work encodes each sub-problem as a complex-valued Gram matrix $G$, not a data matrix $X$. This requires us to express the complex-SVR in the dual variables instead.

By direct computation, the complex-valued Gram matrix $G$ obtained from the complex-valued $X$ is

$$
G = XX^H = X_R^2 + X_I^2 + i.\left( X_I X_R^T - X_R X_I^T \right) = G_R + i.G_I,
\tag{A.27}
$$

where we used $i$ to denote a pure imaginary unity.

Again by direct computation, the real-valued Gram matrix $G'$ obtained from the equivalent Eq. A.26 is

$$
G' = X'X'^T = \left[ \begin{array}{cc} X_R^2 + X_I^2 & X_R X_I^T - X_I X_R^T \\ X_I X_R^T - X_R X_I^T & X_R^2 + X_I^2 \end{array} \right]
\tag{A.28}
$$

Comparing Eq. A.27 to A.28, we see that

$$
G' = \left[ \begin{array}{cc} G_R & G_I^T \\ G_I & G_R \end{array} \right],
\tag{A.29}
$$

and thus we can use Eq. A.29 to express the complex-valued $G$ of a complex-valued SVR with an equivalent real-valued $G$. A simple implementation is shown in Algorithm 3 (Appendix B).

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

[Supplementary Material 2]

# Fast Training of Pose Detectors in the Fourier Domain

**João F. Henriques**     **Pedro Martins**     **Rui Caseiro**     **Jorge Batista**

Institute of Systems and Robotics
University of Coimbra
{henriques,pedromartins,ruicaseiro,batista}@isr.uc.pt

## B   MATLAB code and additional figures

This appendix contains code and additional figures that were not included in the main paper to meet length requirements.

---

**Algorithm 1** MATLAB code for fast Fourier training of several pose detectors. $X$ contains sample groups, with $s$ transformed samples each (e.g. rotations of the same object). Note that the transformation should be cyclic. The samples within a group must also have the same Euclidean norm.

---

Inputs:
- X ($m \times n \times s$ data matrix with $m$ features, $n$ sample groups and $s$ transformations)
- Y ($n \times s$ labels matrix)
- $\lambda$ (scalar, regularization parameter)
- regression (a dual complex-valued regression algorithm – Alg. 2 or Alg. 3)

Output:
- W ($m \times s$ weights matrix – one pose detector per column)

```matlab
g = zeros(n, n, s);
for p = 1:s
  g(:,:,p) = X(:,:,1).' * X(:,:,p);            %Equation 13
end
X = fft(X, [], 3);
g = fft(g, [], 3);
Y = conj(fft(Y, [], 2));
for f = 1:s                                    %Equation 12
  W(:,f) = X(:,:,f) * regression(g(:,:,f), Y(:,f), lambda);
end
W = real(ifft(W, [], 2));
```

---

**Algorithm 2** Complex-valued Dual Ridge Regression (used with Alg. 1).

---

```matlab
function alphas = regression(G, y, lambda)
  alphas = (G + lambda * eye(size(G,1))) \ y;  %Equation 4
end
```

---

**Algorithm 3** Complex-valued Dual Support Vector Regression (can replace `regression` in Alg. 1). The `real_svr` function can be any off-the-shelf SVR solver that accepts a "custom kernel matrix" (Gram matrix), e.g. `libsvm`.[1]

```
function alphas = svr(G, y, lambda)
  G = [real(G), imag(G).'; imag(G), real(G)];        %Equation A.29
  y = [real(y); imag(y)];
  alphas = real_svr(G, y, lambda);
  alphas = alphas(1:end/2) + 1i * alphas(end/2+1:end);
end
```

Figure 1: Visualization of HOG templates, obtained with Fourier training and RR, for the Satellite dataset. Each template represents a distinct pose across planar rotations. Only the templates from 0º to 90º are shown. Positive weights are shown on the left, negative weights on the right. The tight-fitting bounding box is also displayed as a yellow line. Note that rotated HOG templates cannot be obtained from a single template simply by applying a rotation to the cells – the gradient bins must also adjust their orientation correctly. For more complicated transformations and features, this cannot be done in closed form. The implicit transformation model allows us to bypass this issue, and obtain correct gradient orientations.

Figure 2: Visualization of HOG templates, obtained with Fourier training and RR, for the KITTI dataset. Each template represents a distinct pose across out-of-plane rotations (along the vertical axis). Positive weights are shown on the left, negative weights on the right. The tight-fitting bounding box is also displayed as a yellow line.

Figure 3: Visualization of HOG templates, obtained with Fourier training and RR, for the TUD-Campus/TUD-Crossing dataset. Each template represents a distinct pose across a pedestrian's walk cycle. Positive weights are shown on the left, negative weights on the right. The tight-fitting bounding box is also displayed as a yellow line.

## Footnotes

[1] `http://www.csie.ntu.edu.tw/~cjlin/libsvm/`