[Reviews · NeurIPS 2014]

Submitted by Assigned_Reviewer_18

This work presents a method to efficiently train object detectors in the presence of geometric transformations that can be represented as vector-matrix multiplications. This has recently been developed for the case of translation transformations ([1,8,14]) but has not been too obvious for other transformations, such as rotations, let alone non-rigid deformations.

The authors propose to adapt the Fourier-based method that originally allowed the development of efficient algorithms for the case of translations so that we can now deal with rotations and other `cyclic' signal transformations (e.g. the walking pattern of a pedestrian). The condition for this to hold is that the transformation is norm-preserving, can be represented as a matrix multiplication, x_transformed = Q x, has an inverse Q^{-1} = Q^T and for some s Q^s = I.

The starting point for the previous works was the fact that the 'data matrix' obtained by 'stacking' together all translated versions of a signal is a circulant NxN matrix, where N is the length of the signal, and as such can be diagonalized using the discrete harmonic basis (or, Discrete Fourier Transform-DFT matrix), Eq. 3, and reference [5].
The authors note that in the case of general transforms (not translations) the 'data matrix' may not be circulant in general - but as long as the conditions above hold, the Gram matrix will be. As such, one can exploit the DFT-based diagonalization to efficiently train a detector.
Using these, the authors obtain closed-form solutions for ridge regression that only involve inversions of diagonal Gram matrixes - and present also a nice extension for the simultaneous training of multiple pose detectors.

Turning to applying this idea to visual data, we are presented with two different methods for constructing the Gram matrix, one involving an analytic construction of the transformation matrix Q, and another being more `data-driven', where there is in principle no underlying linear transformation, but rather a cyclic pattern, as in the walking cycle of pedestrians, or even out-of-plane rotations of cars.

Result-wise the authors demonstrate remarkable acceleration in terms of training time, albeit with an occasional loss in performance, even when comparing to a ridge regression baseline.

This work presents a method to efficiently train object detectors in the presence of geometric transformations that can be represented as vector-matrix multiplications. This has recently been developed for the case of translation transformations ([1,8,14]) but has not been too obvious for other transformations, such as rotations, let alone non-rigid deformations.

The authors propose to adapt the Fourier-based method that originally allowed the development of efficient algorithms for the case of translations so that we can now deal with rotations and other `cyclic' signal transformations (e.g. the walking pattern of a pedestrian). The condition for this to hold is that the transformation is norm-preserving, can be represented as a matrix multiplication, x_transformed = Q x, has an inverse Q^{-1} = Q^T and for some s Q^s = I.

The starting point for the previous works was the fact that the 'data matrix' obtained by 'stacking' together all translated versions of a signal is a circulant NxN matrix, where N is the length of the signal, and as such can be diagonalized using the discrete harmonic basis (or, Discrete Fourier Transform-DFT matrix), Eq. 3, and reference [5].
The authors note that in the case of general transforms (not translations) the 'data matrix' may not be circulant in general - but as long as the conditions above hold, the Gram matrix will be. As such, one can exploit the DFT-based diagonalization to efficiently train a detector.
Using these, the authors obtain closed-form solutions for ridge regression that only involve inversions of diagonal Gram matrixes - and present also a nice extension for the simultaneous training of multiple pose detectors.

Turning to applying this idea to visual data, we are presented with two different methods for constructing the Gram matrix, one involving an analytic construction of the transformation matrix Q, and another being more `data-driven', where there is in principle no underlying linear transformation, but rather a cyclic pattern, as in the walking cycle of pedestrians, or even out-of-plane rotations of cars.

Result-wise the authors demonstrate remarkable acceleration in terms of training time, albeit with an occasional loss in performance, even when comparing to a ridge regression baseline.

There is in my understanding a substantial amount of originality in this paper, proposing to combine recent advances in fast learning of object detectors in the presence of translations, with more challenging transformations. Some 'out of the box' thinking was required here, as it is most common to deal with rotations through polar coordinates - which would make everything harder; it turns out that by using the Gram matrix we are back to the setting of [14], which is a not-too-evident result; so I was very interested when originally reading the paper.

Still, the paper leaves more to be desired to make the paper significant to the learning/vision community.
The most obvious critique is that the detection results are not compared to a stronger baseline, e.g. an SVM-based detector, so we do not know to what extent the presented method will be practically useful. Along the same lines, the authors could build a pose-invariant detector out of their system (by using the max over poses, as they discuss) and compare it to the methods that are currently out there for detection in the presence of pose variation (e.g. [24] or [c] below).

It is also disappointing to see that there is no discussion about how this method could be extended to other problems, beyond ridge regression. In [14], and (a) below, a combination of similar ideas with the hinge loss has been pursued; it should at least be discussed, and ideally tried out. This is, in my opinion, the most important development that would be needed to make this paper have high impact. As it stands, the work would be of use only to practitioners who want a 'quick and dirty' detector, which may be only as good as one trained with ridge regression.

In particular it is not true that ridge regression (+ whitening) performs similar to SVMs for detection; it merely serves as an 'ok' proxy. This was made more pronounced in the DPM training work of R. Girshick, J. Malik 'Training Deformable Part Models with Decorrelated Features' in ICCV 2013 - there was quite a big difference in performance with and without the final SVM training stage.

I found it also disappointing that the presentation is a bit blurred, leaves several gaps to be filled in by the reader's imagination. I provide below a (partial) list of points where this was the case.

Eq. 3:
The DFT matrix F is never defined; so even if the reader has a signal processing background, one still needs to think about the period of the DFT (is it s or m?) whether F^H is the inverse of forward DFT (I understand it is the inverse) and what the normalization factors are; a single line equation would have made all of this clear.

l. 157-158: 'the matrices involved in a learning problem become diagonal, which drastically reduces the needed computations': There are many learning problems, and many ways in which matrices may be involved. Unless the authors pin down their optimization problem, show how matrices are involved in it, and clarify to what extent the computations become reduced, this is a vague statement.

l. 182: 'structure of the covariance matrix is unknown': we do know the structure of the covariance matrix (we have an equation for it in l. 174 - and it is diagonalizable); we simply cannot express it in terms of the DFT matrix.

l. 182: 'the diagonalization trick ... offers a way out, by computing the Gram matrix, and solving the dual problem'->
The Gram matrix can be computed anyway, and the dual problem can be solved irrespectively of the diagonalization trick. So maybe the authors should be a bit more precise, stating e.g.
'The diagonalization technique allows us to express the Gram matrix in a form which makes an analytical solution of the dual problem possible.'

l. 190: 'a regularized risk': The authors have to provide the form of the regularized risk. It is very common indeed, but presentation-wise it cannot be that the single most important thing (the optimization objective) is never written down. Strictly speaking the paper does not make sense from here on, since there are no 'primal', 'dual' objectives provided anywhere, but the authors keep referring to them. Lack of space is not a real excuse, several more verbose parts of the text can easily become condensed.

l. 191: why should we have 's' dual variables? Where do they show up?
In general, it is bad practice to leave all technical details to other references [5]/[23], since this practically requires that the reader reads three papers rather than one. A paper needs to be as self-contained as possible, and we are talking about things that can be explained with a couple of equations.

l. 198-205: this could be placed right after theorem 1. Here it breaks the presentation's flow

l. 209: the reason why Eq. 6 follows from applying the circulant decomposition to G is not entirely obvious: to a casual reader it may seem as if you were saying F(a/b) = F(a)/F(b), with F being the Fourier transform, which does not make too much sense.
The result is correct indeed (it is a one line proof), but some hand-holding would help. Stating a few central properties of circulant matrices in Section 2.2 would be useful.

Section 4.1: this is, in my understanding, the same story as [14] (they also pass through the gram matrix in their optimization, and finally end up solving many simpler problems). If this is not the case, the difference should be clarified, it it is, it should be stated.

Section 5.1: from the style of the presentation, it is not too clear what exactly the authors do; we read about 'possibilities' (l.299,l.322), but it would be better if the authors state in advance that we use 'such and such a method for this experiment because of this reason' and then go on to describe what the method consists in.

Section 5.3: I could not understand this section. Apparently the authors are describing a technical problem that they faced, but they describe it in a somehow esoteric way : 'we have no principled way of obtaining negative samples for different poses for the same transformation ... lacking a definitive answer a simple strategy is .. - we can further optimize this by realizing that..'
I understand that it is about negative samples not coming with a pose annotation - but I would guess that if one knows the transformation (e.g. rotation) all one has to do is apply all transformations to each negative sample, and label all transformed samples as negatives - so I do not see what is special about negative samples.
I cannot see why the samples are constant along the DFT dimension. The labels may be (all zeros), but the features will be changing if we transform the image.

l. 371: rephrasing is needed (break into sentences, describing the setup: Pascal criterion assigns ground truth object to hypothesis; pose error is measured with respect to that object, as #i/#poses where #i is discretized pose difference)

Table 2: what do you mean by 'with calibration'? Was the proposed method not calibrated?

Appendix: to prove that the data matrix 'will not be circulant in general', we would also need a counterexample for the case of non-translations (I understand that there are plenty - but providing one is a formal requirement).

One work that seems to me relevant (in the sense that it is learning the transformation Q) is
[a] Transformation Equivariant Boltzmann Machines, Jyri J. Kivinen and Christopher K. I. Williams, ICANN 2011,

The authors should also cite a precursor to the more recent works [1,14]:
[b] Maximum Margin Correlation Filter: A New Approach for Simultaneous Localization and Classification
Andres Rodriguez, Vishnu Naresh Boddeti, B.V.K Vijaya Kumar and Abhijit Mahalanobis
IEEE Transactions on Image Processing 2013

Finally, this is one of the works working on detecting rotated cars, the authors could compare to it (or [28])

[c] Andrea Vedaldi, Matthew B. Blaschko, Andrew Zisserman: Learning equivariant structured output SVM regressors. ICCV 2011

Summary: The paper contains an interesting idea, but is somehow inconclusive (it is hard to tell whether this idea would be broadly useful). I found it an interesting read, but I think it requires more work.

Submitted by Assigned_Reviewer_31

The paper proposes a fast method to train pose detectors/classifiers in the Fourier domain. The idea of training pose detectors in the Fourier domain has been extensively used for cyclic translations, since this transformation can be diagonalised in the Fourier Domain, thus dramatically reducing the computational cost. However, up to now this idea has not been extended to more general transformations. In this paper, the authors show that the “Fourier trick” can be generalised to other cyclic transformations. This is a significant contribution, since the proposed framework allows to consider a large class of image transformations that are particularly useful for pose detection, with the benefit of training acceleration provided by the DFT. The authors show that within their proposed approach, image transformations do not necessarily need to be characterised analytically since they become implicit. Therefore, classifiers can be trained not only for virtually generated image transformations, but also for natural datasets with pose annotations.

The framework proposed by the authors also allows training of multiple pose classifiers simultaneously. This is another major improvement with respect to previous work, since training times are dramatically reduced with respect to current approaches that only allow training classifiers for different poses independently.

The article is very well written and organised. A brief but complete overview of relevant related work that clearly points out the advantages of the proposed approach is first presented. Then, the authors show how the can set up the pose classification problem in terms of circulant matrices, thanks to Theorem 1, by solving the dual problem. For that they propose (as other previous work do) to use Ridge Regression, here in the dual domain, and then retrieve the primal solution, leading to a simple closed formula, that can be easily computed, leading to an impressive computational cost reduction with respect to previous work. The framework is presented in increasing complexity, starting from the training of a single classifier for a single cyclic orthogonal transformation of a single image (Sect. 3.1), then extending it to multiple classifiers (one for each pose) trained simultaneously but still from a single image (Sect. 3.2), and finally extending it to the transformation of multiple images (Sect. 4). This final extension benefits from the fact that different Fourier components can be computed independently, making it naturally suited for parallel implementation. In Sect. 5 the fact that the transformation only appears implicitly in the proposed approach is exploited to enlarge the class of considered transformations (including natural transformations, e.g. non-rigid, using pose annotations).

Experiments on three very different settings using standard databases show the suitability of the approach, specially regarding training acceleration.

To sum up, this is a very interesting paper, that presents original contributions, and that has the beauty of being based on extremely simple ideas and principles that lead extremely powerful results. I recommend the paper to be accepted with a few minor corrections:

- the meaning of the acronym HOG is never given (Histogram of Oriented Gradients)
- A discussion on how the regularisation parameter lambda is set is missing
- Section 5.3 (Untransformed negative samples) is not clear; please rewrite with more details it and make an effort to make it more clear.
Summary: The paper presents several original contributions to the widely used approach to accelerate training of pose classifiers in the Fourier domain. More specifically, the proposed approach allows to: (i) take into account much more general transformations than the cyclic translations used in previous works; (ii) train multiple classifiers for different poses simultaneously, with no significant increase of computational cost w.r.t. training a single classifier. The paper is clear and really well written. I strongly recommend acceptance of this paper.

Submitted by Assigned_Reviewer_41

This presents a new method for accelerating object detection training stage. The main idea is aligning all training image samples to a fixed position and orientations, so that learning a classifier can be performed with much less degree of freedom. The idea is made based on the observation that most image detection datasets contain objects whose appearance is very similar to each other up to simple image geometric transformations. The actual alignment of feature vectors is performed in Fourier Domain, since rotation, translation, or other geometric transformations can be easily captured in FFT representation. The author proved that if the transformation of training images are norm-preserving and cyclic, the training can be done in a very efficient fashion. Experiments in a few dataset show that the proposed method significantly reduces the feature training time while preserving similar detection accuracy, compared to a full-fledged training approach. The paper's writing is good and easy to understand. However, the paper should also address the following issues to fully demonstrate the advantage of the propose method:
- Time and accuracy evaluation with more datasets especially PASCAL. Since the paper makes a strong assumption that training objects are similar only up to geometric transformation, whether or not this assumption hurts the detection accuracy can be only validated with much more extensive experiments.
- Comparing 3D pose detection result with object detectors designed for 3D pose estimation, such as Estimating the Aspect Layout of Object Categories by Xiang et al.
Summary: This paper presents a new idea to reduce the complexity in training object detectors. The mathematics behind the method is well studied and presented. However, the paper lacks convincing experiment results to show the accuracy trade-off from increasing training speed is minor and worthwhile.
Author Feedback
Author rebuttal: We thank all reviewers for their helpful comments, and are glad that they (in particular R1 and R2) have enjoyed our paper. We are extremely encouraged that they consider it has a "substantial amount of originality" and "out of the box thinking" (R1), and the "beauty of being based on extremely simple ideas and principles that lead to extremely powerful results" (R2).

Although R2 and R3 consider the article to be very well written, there is always room for improvement and R1 makes many important suggestions. We want our paper to realize its full potential, and as such we will integrate them in the final version.

R1:
Our choice of Ridge Regression for both our method and the baseline is seen as a weakness by R1. The source of this impression can be traced to [12], which presents whitening/LDA as a quick-and-dirty alternative to SVM, with large losses on Inria Pedestrians (from 79.6% AP to 75.1%). However, we must stress that LDA is *not* the same as Fourier-space Ridge Regression techniques [1, 8, 14]. One key difference is that the authors of [12] estimate and invert a 10000x10000 covariance matrix, which is reported to be rank-deficient, while [1, 8, 14] estimate several *independent* 36x36 covariance matrices, which can be done with much greater accuracy. Multiple practitioners have pointed out that these techniques perform better than SVM without bootstrapping [1, 8, 14], and SVM only achieves marginally better performance (<1%) after many bootstrapping rounds. As an example, the publicly available implementation of [14] achieves 80.7% AP on Inria with Ridge Regression, which is very far from the figure of 75.1% given in [12], for LDA. To the credit of [12], it is a precursor and an inspiration that shaped these later techniques.

In conclusion, our baseline is indeed strong, but we also have results with SVM that corroborate these previous works. We will include them to dispel any doubts.

R1 also makes a nice suggestion, that we explore algorithms other than Ridge Regression. From [14], extensions of our method for Support Vector Regression (SVR) and others can be obtained easily. However, rather than go for full generality, we opted to keep the discussion focused, since the solution in Section 4 already deals with many factors simultaneously (arbitrary transformations + multiple base samples + multiple output classifiers). We hope this is understandable. We will briefly discuss it in the paper, and include the (straightforward but tedious) SVR derivation and experimental results either as an appendix or a separate technical report (SVR scores only about 1-2% higher AP).

Concerning calibration, our method does not require it (l.366).

R2:
We realize that the paragraph of Section 5.3 is probably the least clear and we will reword it accordingly. The problem at hand boils down to the lack of pose annotations in negative samples. For planar rotation we can easily obtain rotated negative samples with random poses. However, the same operation with walk cycles of pedestrians is not defined. How do we advance the walk cycle of a non-pedestrian? As a pragmatic solution, we consider that transformations of this type have no effect on negative samples. This allows us to solve this quandary. Incidentally, their DFT in pose-space turns out to be 0 everywhere, except for the DC value, which allows massive computational savings. We hope this also answers a question by R1.

R3:
Unfortunately, there is a bit of a misunderstanding of the main idea of our paper, which we hope we can clarify. We do not perform alignment of samples to a canonical pose in the Fourier domain. Instead, we provide a fast solution for the case of simultaneously training with *all* poses of the given samples. There are no assumptions of 3D geometry, as we demonstrate with the non-rigid walk cycle experiment, and the pose transformation is learned within a monolithic HOG template. Our method is complementary to the suggested work on 3D aspect layouts, and can be used to train the individual part templates for a range of viewpoints (which would remove the need for rectification and allow non-planar aspect parts). We will make this distinction in the final version and include the suggested reference.

About Pascal VOC, it is not the case that any single transformation is responsible for most of the appearance variability. Our method is not directly applicable to such a scenario without handling multiple transformation types simultaneously, a non-trivial problem that will be considered in future work. However, there are a multitude of computer vision tasks where handling a single transformation is extremely useful -- we have demonstrated results in 3 very different settings (pedestrian walk cycles, satellite images, and the azimuth of cars in street scenes). This should take care of any doubts of broad applicability. We must also emphasize that the KITTI dataset (http://www.cvlibs.net/datasets/kitti/) is large-scale, very realistic and of high significance for a real-world application (autonomous driving).